# Unmasking the Hidden Meaning: Bridging Implicit and Explicit Hate Speech Embedding Representations

**Nicolas Ocampo** and **Elena Cabrio** and **Serena Villata**

Universite Côte d'Azur, CNRS, Inria, I3S, France

{nicolas-benjamin.ocampo,elena.cabrio,serena.villata}@univ-cotedazur.fr

## Abstract

Research on automatic hate speech (HS) detection has mainly focused on identifying explicit forms of hateful expressions on user-generated content. Recently, a few works have started to investigate methods to address more implicit and subtle abusive content. However, despite these efforts, automated systems still struggle to correctly recognize implicit and more veiled forms of HS. As these systems heavily rely on proper textual representations for classification, it is crucial to investigate the differences in embedding implicit and explicit messages. Our contribution to address this challenging task is fourfold. First, we present a comparative analysis of transformer-based models, evaluating their performance across five datasets containing implicit HS messages. Second, we examine the embedding representations of implicit messages across different targets, gaining insight into how veiled cases are encoded. Third, we compare and link explicit and implicit hateful messages across these datasets through their targets, enforcing the relation between explicitness and implicitness and obtaining more meaningful embedding representations. Lastly, we show how these newer representation maintains high performance on HS labels, while improving classification in borderline cases.

## 1 Introduction

The proliferation of hate speech (HS) on social media platforms has become a pressing concern in online social communities. While significant progress has been made in the development of HS detection methods, current SOTA models focus on detecting explicit HS, leaving implicit hate cases undetected (ElSherief et al., 2021; Ocampo et al., 2023). This issue is aggravated by the sheer volume of implicit hate speech content being spread across various online platforms, necessitating automated approaches to detect them effectively.

Implicit HS detection poses unique challenges compared to its explicit counterpart: it contains coded, ambiguous or indirect language that does not immediately denote hate, but still disparages a person or a group based on protected characteristics such as race, gender, cultural identity, or religion (e.g., *"I think it is a bit late to think to look after the safety and the future of white people in South-Africa"* - the White Supremacy Forum Dataset (de Gibert et al., 2018)). The performance of current HS systems heavily relies on how coded language is represented and how well classifiers can capture the underlying semantic meaning of messages through embeddings. Hence, obtaining better text representation becomes crucial in effectively identifying implicit HS messages.

In this direction, the goal of our work is to bridge the gap between explicit and implicit messages, aiming to enhance the embedding representations of SOTA models. Our contribution is fourfold: *i)* We analyze the embedding representations of five benchmark datasets with veiled hateful content, examining the levels of explicitness and implicitness, through cross-evaluation using state-of-the-art transformer models. *ii)* We examine the embedding representations of implicit messages across different target groups. Through this analysis, we gain insights into how implicit HS messages are encoded based on their target groups. *iii)* We propose a novel approach to link explicit and implicit HS messages in the representation space. *iv)* We illustrate that the newer representation space preserves strong efficacy for HS labels, while also refining classification in borderline instances. Using contrastive learning techniques (Gunel et al., 2020; Rethmeier and Augenstein, 2021; Kim et al., 2022; Tian et al., 2020), we aim to push explicit and implicit messages effectively enforcing the uncovered relation between these two notions and thereby obtaining more meaningful representations than those obtained through fine-tuning learning methods.[1]

---

[1] The accompanying software can be found at: https://github.com/benjaminocampo/bridging_ie_hs_embs.

*NOTE: This paper contains examples of language which may be offensive to some readers. They do not represent the views of the authors.*

## 2 Related Work

HS detection has been extensively studied by the research community providing multiple resources, such as lexicons (Wiegand et al., 2018; Bassignana et al., 2018), datasets (Zampieri et al., 2019; Basile et al., 2019; Davidson et al., 2017; Founta et al., 2018), and supervised methods (Park and Fung, 2017; Gambäck and Sikdar, 2017; Wang et al., 2020; Lee et al., 2019) (for a survey, see (Poletto et al., 2021)). These studies provide a solid starting point to examine the problem of abusive language, especially in social media messages. Lately, there has been growing interest in the detection of implicit HS, which provides additional challenges. Datasets specifically designed for implicit HS (Ocampo et al., 2023; Hartvigsen et al., 2022; ElSherief et al., 2021; Vidgen et al., 2021; Sap et al., 2020), more solid veiled detectors (Han and Tsvetkov, 2020), guided augmentation strategies (Nejadgholi et al., 2022; Roychowdhury and Gupta, 2023), and theoretical analysis (Jurgens et al., 2019; Waseem et al., 2017; Wiegand et al., 2021) have been recently proposed to advance in this direction.

However, little attention has been dedicated to effectively represent implicit messages through embeddings on these benchmarks. Embeddings play a crucial role in the performance of classifiers (Pavlopoulos et al., 2017; Kshirsagar et al., 2018; Ocampo et al., 2023), yet their application to capture the implicit nature of HS has been under-investigated. In this direction, (Kim et al., 2022) tackles cross-dataset underperforming issues on HS classifiers and proposes a contrastive learning method that encodes a hateful post and its corresponding implication close in representation space, closely depending on the annotated implications and without contra positioning explicitness with implicitness. (Bourgeade et al., 2023) captures topic-generic and topic-specific knowledge when trained on different data to improve generalization.

## 3 Implicit and Explicit HS Embeddings

### 3.1 Research Questions

We will focus on the behavior of SOTA models in cross-evaluation settings, specifically on datasets containing implicit hate. The study explores the models' behavior on different HS classes, including both explicit and implicit hate. In particular, we target the following research questions (RQ):

**RQ1**: How do the models' embeddings capture the HS classes? Are explicit and implicit hateful messages encoded differently across different datasets? What is the extent of this variation?

**RQ2**: Does grouping the test sets by target result in similar encoding patterns for explicit HS and distinct encoding patterns for implicit HS in the embeddings? RQ2 builds upon the analysis conducted in RQ1, but with a focus on target groups.

**RQ3**: Can we link and bring explicit and implicit embedding representations closer together within the learned embedding space through their target groups?

**RQ4**: How do these newer embedding representations capture HS classes in comparison with RQ1?

### 3.2 Datasets

We carried out our analysis on the following standard datasets, containing implicit HS messages: Implicit Subtle Hate (ISHate) (Ocampo et al., 2023), Social Bias Inference (SBIC) (Sap et al., 2020), Implicit Hate Corpus (IHC) (ElSherief et al., 2021), Dynahate (DYNA) (Vidgen et al., 2021), and Toxigen (TOX) (Hartvigsen et al., 2022). We ensured that the definitions of HS were consistent across the datasets. Specifically, for the SBIC dataset, messages are considered as HS if labeled as offensive and target a specific group. As for the explicit-implicit HS labeling across all datasets, the provided implicit labels are used for IHC and ISHate datasets. For SBIC, DYNA, and TOX, we computed the percentage of HS implicit messages as the ones where none of the words of the Google profanity words resource was present[2]. The datasets were divided into train, dev, and test sets. Existing dataset splits were retained, while datasets without predefined splits were divided using a stratified splitting method with a 60% train, 20% dev, and 20% test ratio. Table 4 in Appendix shows the percentage of implicit/explicit instances per dataset.

### 3.3 Experimental Settings

Concerning our research questions (Section 3.1), to answer to **RQ1** we performed fine-tuning on two state-of-the-art models commonly used for HS detection: BERT and HateBERT (Caselli et al., 2021). Both models were fine-tuned on each dataset using

---

[2]List of swear words banned by Google: `https://github.com/RobertJGabriel/Google-profanity-words`

a two-label classification approach, distinguishing between non-HS and HS messages. To ensure robustness and account for randomness in the training process, we repeated the fine-tuning procedure five times, each time employing a different random seed. This allowed us to evaluate the performance of the models consistency. To assess the performance of the models, we cross-evaluate the benchmarks calculating the average F1-score across all fine-tuning runs. Additionally, we calculated the standard deviation to quantify the variability in performance observed across the different runs. Finally, we calculated the embeddings of these models using TSNE highlighting how explicit and implicit messages were encoded. We used the base versions size of these models with batch size of 32, weight decay of 0.01, 4 epochs, and a learning rate of 2e-5. As for TSNE, we use perplexity of 30, and 1000 maximum iterations for convergence.

For **RQ2**, we grouped the embeddings per target in the plots. To ensure consistency across datasets, we standardized the target names, addressing label variations, e.g., we resolved differences like "asian" and "asian people" by using a unified label. Moreover, when a message targeted multiple offensive groups (e.g. Asians and Migrants), we selected the label corresponding to the predominant target (among MUSLIMS, WOMEN, JEWS, LGBTQ+, BLACK PEOPLE, WHITE PEOPLE, IMMIGRANTS, ASIAN, and DISEASE).

For **RQ3**, we aim to validate the potential linkage between explicit and implicit messages through their target groups. To achieve this, we employ contrastive learning techniques on the pre-trained and fine-tuned models. Contrastive learning involves defining pairs of positive and negative samples and training the model using a modified loss function. In our experimental settings, we designate pairs of implicit and explicit messages with the same target as positive samples. For each implicit ones, a randomly selected explicit message with the same target is paired. In cases where they are unavailable or when the implicit instance lacks a target label, we randomly assign any explicit message. Negative samples consist of pairs of HS and Non-HS instances. For every Non-HS instance, one HS instance is randomly selected. Using contrastive learning facilitates the training process by pushing positive pairs closer together while pushing negative pairs further apart within the embedding space.

The contrastive loss is defined as follows:

$$\text{loss\_cont} = mean\left((1-l) \cdot s^2 + l \cdot (max(0, m-s))^2\right) \quad (1)$$

Where $l$ represents the label pair (1 for positive pairs, 0 for negative pairs), $s$ is the cosine similarity between paired messages, and $m$ is the margin hyper-parameter. For classification, the cross-entropy loss is:

$$\text{loss\_clf} = -\sum_{i=0}^{N-1} \left(g_i log(p_i) + (1-g_i)log(1-p_i)\right) \quad (2)$$

Where $g$ is the gold label (labels of the dataset on which the model is fine-tuned) and $p$ is the prediction. The final loss is:

$$\text{total\_loss} = \text{loss\_cont} + \text{loss\_clf} \quad (3)$$

By combining them, we optimize both the model's understanding of embeddings and classification.

For **RQ4**, we fine-tuned both BERT and Hate-BERT using our enhanced embeddings (same settings of our initial RQs). Additionally, to gain more targeted diagnostic insights, models' accuracy was evaluated on three categories defined on the SBIC dataset (Non-HS, Explicit HS, and Implicit HS), and the HateCheck dataset (Röttger et al., 2021), a suite of functional tests for HS detection models.

### 3.4 Obtained Results and Discussion

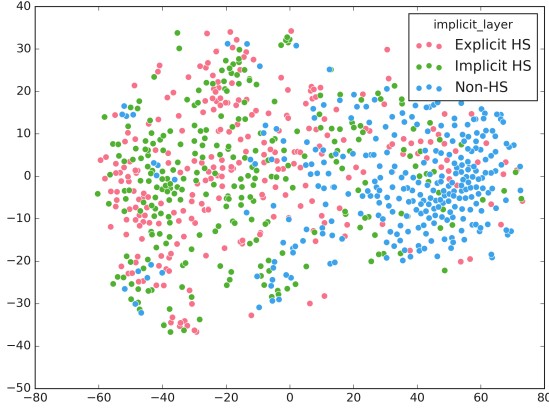

Figure 1: **RQ1**: TSNE embeddings of the SBIC test set using HateBERT fine-tuned on DYNA.

Regarding **RQ1**, Table 1a shows that training and evaluating HateBERT (BERT results can be found in the Appendix) on the same dataset yields

| Train \ Test | IHC | SBIC | DYNA | ISHate | TOX |
|---|---|---|---|---|---|
| IHC | 0,7618 ± 0,0027 | 0,6824 ± 0,0129 | 0,5679 ± 0,0522 | 0,6959 ± 0,0238 | 0,5896 ± 0,0119 |
| SBIC | 0,6385 ± 0,0310 | 0,8632 ± 0,0099 | 0,6355 ± 0,0399 | 0,7522 ± 0,0153 | 0,6998 ± 0,0086 |
| DYNA | 0,6717 ± 0,0310 | 0,7473 ± 0,0357 | 0,7860 ± 0,0111 | 0,7602 ± 0,0025 | 0,7526 ± 0,0075 |
| ISHate | 0,6188 ± 0,0052 | 0,7209 ± 0,0400 | 0,6190 ± 0,0064 | 0,8684 ± 0,0035 | 0,6034 ± 0,0045 |
| TOX | 0,5063 ± 0,0140 | 0,5428 ± 0,0133 | 0,4952 ± 0,0197 | 0,5900 ± 0,0195 | 0,7650 ± 0,0102 |

(a) Cross-evaluation results with HateBERT.

| Train \ Test | IHC | SBIC | DYNA | ISHate | TOX |
|---|---|---|---|---|---|
| IHC | 0,7433 ± 0,0107 | 0,6618 ± 0,0173 | 0,5415 ± 0,0092 | 0,6678 ± 0,0058 | 0,5843 ± 0,0153 |
| SBIC | **0,6593 ± 0,0095** | 0,8620 ± 0,0064 | **0,6591 ± 0,0069** | **0,7583 ± 0,0086** | 0,6742 ± 0,0208 |
| DYNA | 0,6520 ± 0,0102 | 0,7323 ± 0,0116 | 0,7831 ± 0,0030 | 0,7566 ± 0,0135 | 0,7147 ± 0,0075 |
| ISHate | **0,6253 ± 0,0060** | 0,6753 ± 0,0442 | 0,6165 ± 0,0116 | 0,8394 ± 0,0064 | 0,5973 ± 0,0210 |
| TOX | **0,5354 ± 0,0210** | **0,5637 ± 0,0346** | **0,5202 ± 0,0098** | **0,6180 ± 0,0256** | 0,7610 ± 0,0103 |

(b) Cross-evaluation results with Contrastive HateBERT. Bold values indicate improvements compared to Table 1a.

Table 1: HateBERT and Contrastive HateBERT cross-evaluation results with five different run seeds.

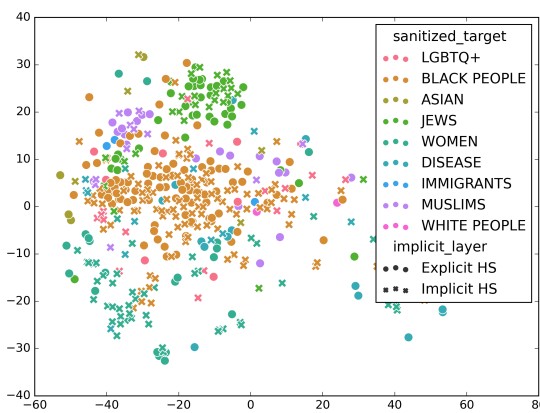

Figure 2: **RQ2**: TSNE embeddings of the SBIC test set using HateBERT fine-tuned on DYNA based on their target groups. Non-HS are excluded from the plot.

better results overall, as could be expected. However, even in cross-evaluation scenarios, reasonable performances are observed. Notably, among the most generalizable models, HateBERT trained on DYNA exhibits better generalization. We therefore selected HateBERT trained on DYNA as the best configuration and we plot the embeddings for the test sets of all the datasets, applying the TSNE algorithm. Figure 1 shows how explicit HS and non-HS messages are encoded with clear separation, resulting in a noticeable distance between them.[3] On the other hand, implicit HS instances tend to be intertwined with both non-HS and explicit HS messages. This pattern holds true across all 5 datasets.

As for the results for **RQ2**, Figure 2 shows ex-

plicit and implicit text representations per target group highlighting how, in general, embeddings of explicit and implicit messages tend to be linked by their target groups in representation spaces. Finally, as for **RQ3**, Figure 3 demonstrates that the embedding representations of explicit and implicit instances starts to overlap across all datasets when using HateBERT trained on DYNA. Additionally, Figure 4 highlights that by leveraging the targets of HS using contrastive learning, explicit and implicit messages exhibit a similar representation.

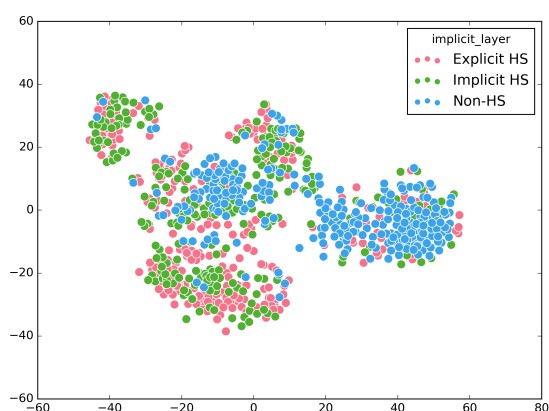

Figure 3: **RQ3**: TSNE embeddings of the SBIC test set using HateBERT fine-tuned on DYNA and linking explicit and implicit instances.

As for **RQ4**, Table 1b illustrates that the novel representation enhances the F1-score for certain datasets, such as SBIC, TOX, and ISHate. Conversely, for other datasets like IHC and DYNA, the performance remains comparable to that of the non-contrastive approach. Table 2 shows higher capability of the contrastive

| Test Case | HateBERT IHC | Contrastive IHC | HateBERT SBIC | Contrastive SBIC | HateBERT DYNA | Contrastive DYNA | HateBERT ISHate | Contrastive ISHate | HateBERT TOX | Contrastive TOX |
|---|---|---|---|---|---|---|---|---|---|---|
| counter_quote_nh | .486 ± .082 | **.499 ± .085** | .276 ± .213 | **.422 ± .114** | .857 ± .048 | **.962 ± .027** | .379 ± .079 | **.524 ± .160** | 0 ± 0 | **.010 ± .009** |
| counter_ref_nh | .576 ± .045 | **.617 ± .086** | .240 ± .168 | **.393 ± .144** | .882 ± .018 | **.902 ± .037** | .387 ± .038 | **.521 ± .187** | .177 ± .043 | **.295 ± .117** |
| ident_pos_nh | .441 ± .020 | .446 ± .136 | .301 ± .076 | **.361 ± .038** | .849 ± .021 | .767 ± .099 | .272 ± .071 | **.350 ± .101** | .470 ± .126 | **.575 ± .133** |
| negate_neg_nh | .502 ± .054 | **.517 ± .134** | .120 ± .060 | **.161 ± .104** | .448 ± .037 | **.486 ± .040** | .164 ± .030 | **.211 ± .106** | .087 ± .036 | **.194 ± .077** |
| profanity_nh | .796 ± .036 | .774 ± .076 | .922 ± .099 | **.994 ± .005** | 1 ± 0 | 1 ± 0 | .992 ± .004 | **.996 ± .005** | .292 ± .101 | **.456 ± .105** |
| slur_homonym_nh | .353 ± .061 | **.413 ± .084** | .393 ± .162 | **.513 ± .099** | .813 ± .030 | .787 ± .073 | .680 ± .056 | **.827 ± .101** | .320 ± .051 | **.473 ± .064** |
| slur_reclaimed_nh | .217 ± .045 | **.277 ± .084** | .472 ± .196 | **.711 ± .117** | .891 ± .018 | .879 ± .044 | .741 ± .070 | **.802 ± .113** | .272 ± .031 | **.346 ± .067** |
| target_group_nh | .710 ± .036 | .700 ± .027 | .623 ± .328 | **.810 ± .072** | .968 ± 0 | **.971 ± .021** | .448 ± .027 | **.561 ± .085** | 0 ± 0 | **.006 ± .014** |
| target_indiv_nh | .538 ± .049 | **.572 ± .104** | .655 ± .451 | **.951 ± .028** | 1 ± 0 | 1 ± 0 | .782 ± .032 | **.809 ± .061** | .003 ± .007 | .003 ± .007 |
| target_obj_nh | .637 ± .042 | .622 ± .105 | .923 ± .086 | **.966 ± .037** | .969 ± .011 | **.985 ± .015** | .735 ± .013 | **.757 ± .073** | .006 ± .008 | **.034 ± .035** |

Table 2: Comparative accuracy performance of HateBERT vs Contrastive HateBERT trained in each dataset and evaluated across various test cases on HateCheck.

| Train | Explicit | Implicit | Non-HS |
|---|---|---|---|
| IHC | 0.8832 | 0.8842 | 0.4659 |
| SBIC | 0.8971 | 0.8440 | 0.8568 |
| DYNA | 0.6071 | 0.6665 | 0.8314 |
| ISHate | 0.5515 | 0.4676 | 0.8768 |
| TOX | 0.7938 | 0.8617 | 0.3439 |

Table 3: Contrastive HateBERT avg accuracy across Explicit, Implicit, and Non-HS (SBIC test set).

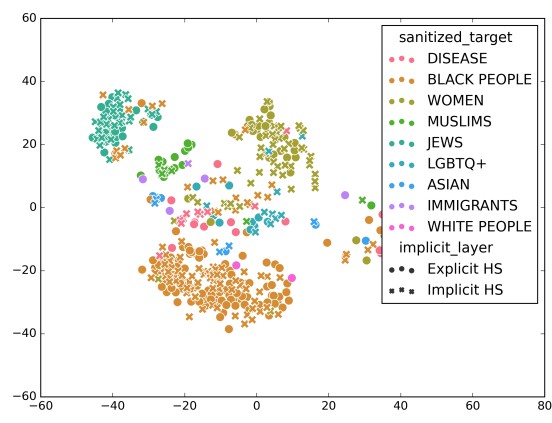

Figure 4: **RQ3**: TSNE embeddings of the SBIC test set using HateBERT fine-tuned on DYNA based on their target groups. Non-HS are excluded from the plot.

HateBERT in accurately classifying challenging Non-HS messages across all five datasets. A significant reduction in false positives is also observed in HateCheck categories such as quoted announcements (counter_quote_nh), direct references (counter_ref_nh), positive identifiers (ident_pos_nh), negated hateful remarks (negate_neg_nh), non-hateful profanity (profanity_nh), reclaimed slurs (slur_reclaimed_nh), homonym slurs (slur_homonym_nh), as well as targeted abuse directed at individuals (target_indiv_nh), objects (target_obj_nh), and non-protected groups (target_group_nh). Additionally, Table 3 indicates that both Explicit and Implicit categories exhibit similarly high accuracy levels, highlighting their nearly indistinguishable impact on the model's aggregate performance. Also, the importance of the Non-HS category is underscored, varying with different training datasets, yet remaining a critical component.

Hence, our experiments emphasize the importance of studying implicit representations, as classical training strategies cannot encode them properly (RQ1). We showed that implicit and explicit messages share a connection conveying similar messages to the same target (RQ2) and how contrastive learning effectively forces that property by bridging explicit and implicit instances through their targets (RQ3), thereby obtaining more meaningful representations that the ones obtained through fine-tuning. Finally, we reduced biases in non-hateful implicit cases often misclassified due to trigger words or nuanced content. Our enhanced method maintains high performance on HS labels while improving classification in borderline cases, proving its robustness and precision (RQ4).

## 4   Conclusions

Our contribution in this study is fourfold: *i)* We studied how models' embeddings capture HS w.r.t. explicitness and implicitness, *ii)* We showed how explicit and implicit HS messages result in similar encodings if grouped by their protected target, *iii)* We analyzed a contrastive learning method to force this property when representing implicit text. We prove our research hypothesis on 5 HS benchmarks, moving a step forward in bridging the gap between explicitness and implicitness, and *iv)* We show how the newer representation space maintains high performance on HS labels while improving classification in borderline cases. In future work, we'll refine contrastive learning, delving into contextual pairing based on other semantic dependencies between explicit and implicit cues, aiming to sharpen nuanced hate speech detection.

## Limitations

In this study, we are aware of some key issues, one of which pertains to the selection of positive and negative samples in contrastive learning. The effectiveness of the algorithm heavily relies on the careful selection of these pairs. While our investigation demonstrates that explicit and implicit messages exhibit a relationship through their target groups across five distinct datasets, it is important to acknowledge that this assumption may not always hold true. Additionally, ensuring a clear separation between non-hateful and HS instances can be challenging due to the heterogeneity of each category.

Moreover, the efficacy of our approach is contingent upon the availability and alignment of target information across the datasets. While target information is commonly provided in benchmark datasets, different datasets may address various protected characteristics. Our approach assumes that there is some degree of overlap in terms of target groups among the selected datasets.

Furthermore, the selection of pairs when linking explicit and implicit messages can vary in terms of the number of combinations. However, it is important to note that as the number of pairs increases, the training requirements tend to grow significantly, resulting in slower training processes. This trade-off between the number of pairs and training efficiency should be carefully considered when implementing the approach.

## Ethics Statement

This paper uses a collection of HS examples extracted from linguistic resources commonly employed for HS detection, ensuring their independence from the authors' personal opinions. The datasets used in this study have been meticulously handled to address privacy concerns associated with user data. While we acknowledge the potential for misuse, we firmly believe that developing robust HS classifiers is essential in combating the proliferation of harmful content. In this regard, our work represents a significant contribution towards this objective and encourages further exploration and investigation within the scientific community.

## Acknowledgements

This work has been supported by the French government, through the 3IA Côte d'Azur Investments in the Future project managed by the National Research Agency (ANR) with the reference number ANR- 19-P3IA-0002.

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

## A  Datasets Statistics and Details

Table 4 displays statistics related to the datasets used in our study, including IHC, SBIC, DYNA, TOX, and ISHate. The table presents the percentage of implicit and explicit instances per dataset, along with their distribution across set partitions, and target groups distribution.

## B  Evaluation results with BERT

In this section we show the evaluation of the BERT and Contrastive BERT models for RQ1 and RQ4 specified in sections 3.1 and 3.3.

Table 5a demonstrates BERT's efficacy in cross-evaluation contexts, mirroring the results seen with HateBERT. Among the models, SBIC stands out, displaying superior generalization capabilities. Conversely, Table 5b illustrates that while models like SBIC, IHC, and TOX reap advantages from contrastive learning, others experience a slight dip in performance, though maintaining an overall high-quality output.

Moving on to Table 6, it's evident that a segment of the enhancement is attributed to the precise categorization of challenging Non-HS messages prevalent across all five datasets. This precision underscores a more conservative and meticulous approach in classifying a message as Hateful.

Finally, Table 7 highlights BERT's consistent performance, boasting high accuracy in handling Non-HS instances for each dataset. This is achieved without compromising the emphasis on discerning between Explicit and Implicit labels, thereby ensuring that the model maintains a balanced focus on varied content nuances.

## C  TSNE embeddings for RQ1, RQ2, and RQ3 in all datasets

This section presents the TSNE results for each research questions RQ1, RQ2, and RQ3, illustrated in Figures 5, 6, 7, and 8. These visualizations are generated from the embeddings captured by Hate-BERT, specifically trained on the DYNA dataset and evaluated on all datasets.

| Datasets | Source | Size | % Implicit | % Explicit | % Hate Class |
|---|---|---|---|---|---|
| IHC | Twitter | 21480 | 86,7017 | 13,2983 | 38,1238 |
| SBIC | Social Media | 144649 | 58,9586 | 41,0414 | 38,9667 |
| DYNA | Human-Machine Adv. | 41144 | 58,0654 | 41,9346 | 53,8961 |
| ISHate | Social Media | 53073 | 71,5686 | 28,4314 | 66,3313 |
| TOX | GPT-3 | 9866 | 42,381 | 57,619 | 48,956 |

Table 4: Comparing toxic language datasets. % Hate Class, % Implicit, and % Explicit are the percent labeled as hate, implicit hate, and explicit hate, respectively.

| Train \ Test | IHC | SBIC | DYNA | ISHate | TOX |
|---|---|---|---|---|---|
| IHC | 0,7625 ± 0,0063 | 0,6891 ± 0,0072 | 0,5511 ± 0,0077 | 0,6824 ± 0,0068 | 0,6074 ± 0,0329 |
| SBIC | 0,6603 ± 0,0301 | 0,8568 ± 0,0092 | 0,6500 ± 0,0310 | 0,7581 ± 0,0167 | 0,6939 ± 0,0086 |
| DYNA | 0,6660 ± 0,0046 | 0,7412 ± 0,0098 | 0,7831 ± 0,0027 | 0,7515 ± 0,0039 | 0,7501 ± 0,0036 |
| ISHate | 0,6214 ± 0,0040 | 0,7480 ± 0,0058 | 0,6279 ± 0,0056 | 0,8635 ± 0,0029 | 0,6012 ± 0,0090 |
| TOX | 0,5455 ± 0,0091 | 0,5855 ± 0,0312 | 0,5193 ± 0,0204 | 0,6140 ± 0,0275 | 0,7824 ± 0,0094 |

(a) Cross-evaluation results with BERT.

| Train \ Test | IHC | SBIC | DYNA | ISHate | TOX |
|---|---|---|---|---|---|
| IHC | 0.7418 ± 0.0066 | 0.6877 ± 0.0157 | 0.5463 ± 0.0183 | 0.6748 ± 0.0117 | **0.6141 ± 0.0100** |
| SBIC | **0.6630 ± 0.0049** | **0.8602 ± 0.0034** | 0.6390 ± 0.0056 | 0.7493 ± 0.0093 | 0.6574 ± 0.0145 |
| DYNA | 0.6431 ± 0.0074 | 0.6973 ± 0.0241 | 0.6979 ± 0.0080 | 0.7414 ± 0.0124 | 0.7001 ± 0.0145 |
| ISHate | 0.6202 ± 0.0156 | 0.6808 ± 0.0398 | 0.6233 ± 0.0078 | 0.8350 ± 0.0097 | 0.5797 ± 0.0206 |
| TOX | **0.5501 ± 0.0396** | 0.5741 ± 0.0410 | 0.5611 ± 0.0264 | 0.6086 ± 0.0349 | 0.7645 ± 0.0103 |

(b) Cross-evaluation results with Contrastive BERT. Bold values indicate improvements compared to Table 5a.

Table 5: BERT and Contrastive BERT cross-evaluation results with five different run seeds.

| Test Case | BERT IHC | Contrastive IHC | BERT SBIC | Contrastive SBIC | BERT DynaHate | Contrastive DynaHate | BERT ISHate | Contrastive ISHate | BERT ToxiGen | Contrastive ToxiGen |
|---|---|---|---|---|---|---|---|---|---|---|
| counter_quote_nh | .297 ± .095 | **.410 ± .070** | .282 ± .143 | **.414 ± .095** | .843 ± .057 | **.870 ± .082** | .319 ± .044 | **.434 ± .050** | .097 ± .051 | **.109 ± .121** |
| counter_ref_nh | .329 ± .055 | **.467 ± .049** | .201 ± .093 | **.264 ± .073** | .848 ± .033 | **.908 ± .018** | .410 ± .033 | **.555 ± .072** | .380 ± .152 | .359 ± .193 |
| ident_pos_nh | .340 ± .039 | **.406 ± .088** | .233 ± .085 | **.361 ± .165** | .846 ± .044 | .808 ± .137 | .317 ± .060 | **.454 ± .046** | .556 ± .130 | **.624 ± .194** |
| negate_neg_nh | .346 ± .054 | **.427 ± .132** | .048 ± .031 | **.128 ± .077** | .406 ± .055 | **.502 ± .062** | .093 ± .022 | **.194 ± .098** | .218 ± .146 | **.262 ± .176** |
| profanity_nh | .810 ± .054 | **.822 ± .052** | .946 ± .115 | **1 ± 0** | 1 ± 0 | 1 ± 0 | 1 ± 0 | 1 ± 0 | .780 ± .203 | .688 ± .168 |
| slur_homonym_nh | .513 ± .045 | .500 ± .120 | .520 ± .090 | .513 ± .112 | .860 ± .037 | **.875 ± .057** | .880 ± .038 | **.953 ± .038** | .667 ± .100 | .667 ± .113 |
| slur_reclaimed_nh | .165 ± .060 | **.215 ± .094** | .398 ± .113 | **.481 ± .081** | .815 ± .045 | .815 ± .035 | .694 ± .064 | **.790 ± .049** | .430 ± .050 | .420 ± .156 |
| target_group_nh | .684 ± .031 | **.716 ± .049** | .732 ± .267 | **.771 ± .070** | .987 ± .013 | **.992 ± .016** | .416 ± .031 | **.529 ± .080** | .048 ± .036 | **.084 ± .119** |
| target_indiv_nh | .557 ± .063 | .498 ± .135 | .797 ± .351 | **.938 ± .038** | 1 ± 0 | 1 ± 0 | .695 ± .028 | **.818 ± .077** | .142 ± .127 | .074 ± .073 |
| target_obj_nh | .677 ± .067 | **.711 ± .111** | .988 ± .028 | .978 ± .018 | 1 ± 0 | 1 ± 0 | .785 ± .031 | **.855 ± .056** | .228 ± .182 | .218 ± .175 |

Table 6: Comparative accuracy performance of BERT vs Contrastive BERT trained in each dataset and evaluated across various test cases on HateCheck.

| Train | Explicit | Implicit | Non-HS |
|---|---|---|---|
| IHC | 0.8754 | 0.8678 | 0.5206 |
| SBIC | 0.8920 | 0.8348 | 0.8607 |
| DYNA | 0.5796 | 0.6113 | 0.8068 |
| ISHate | 0.5542 | 0.4387 | 0.9067 |
| TOX | 0.6892 | 0.7727 | 0.4408 |

Table 7: Contrastive BERT avg accuracy across Explicit, Implicit, and Non-HS (SBIC test set).

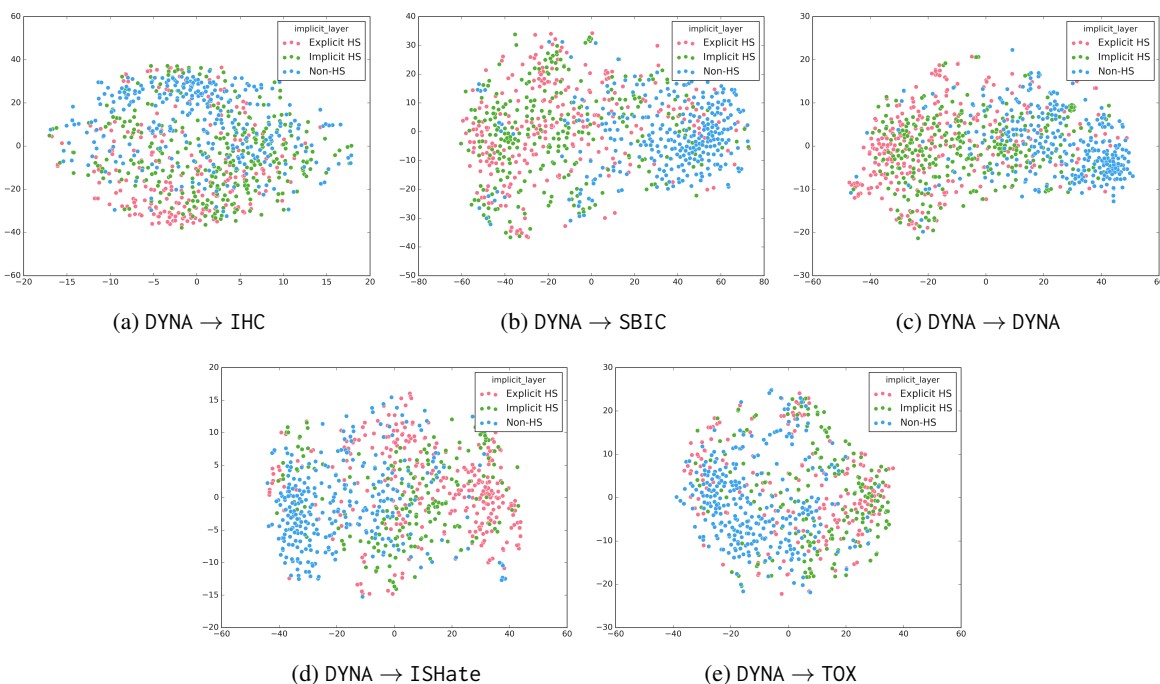

Figure 5: **RQ1**: TSNE embeddings of the test sets for all the datasets using HateBERT fine-tuned on DYNA.

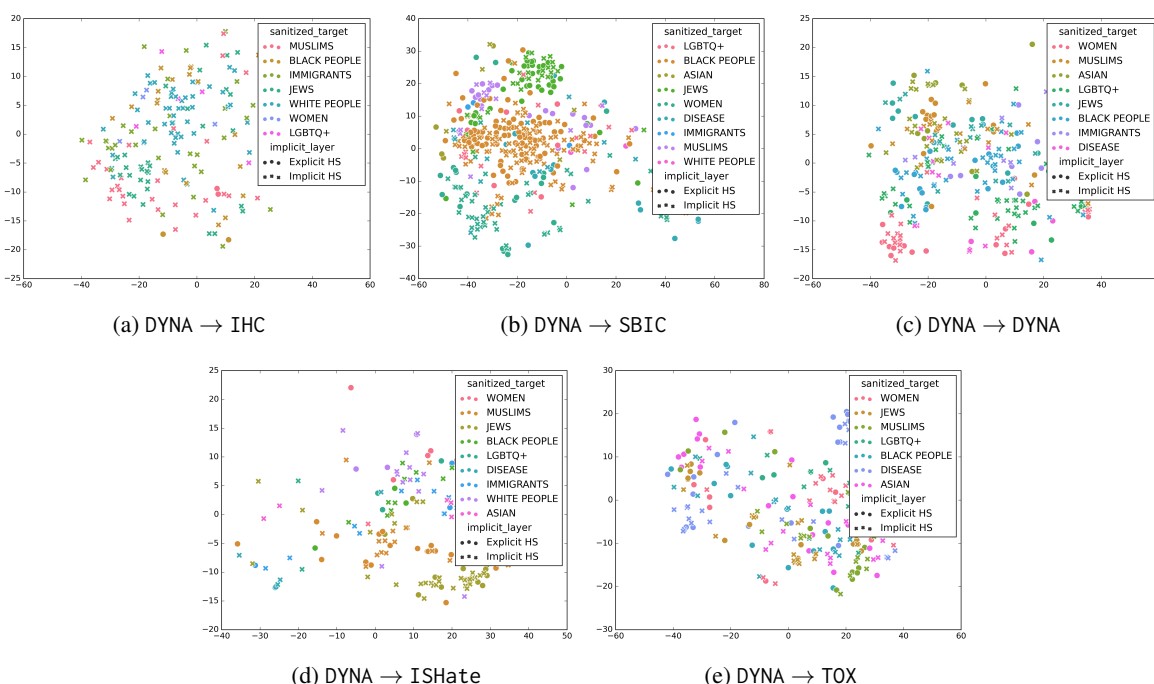

Figure 6: **RQ2**: TSNE embeddings of the test sets for all the datasets using HateBERT fine-tuned on DYNA based on their target groups. Non-hateful instances are excluded from these plots.

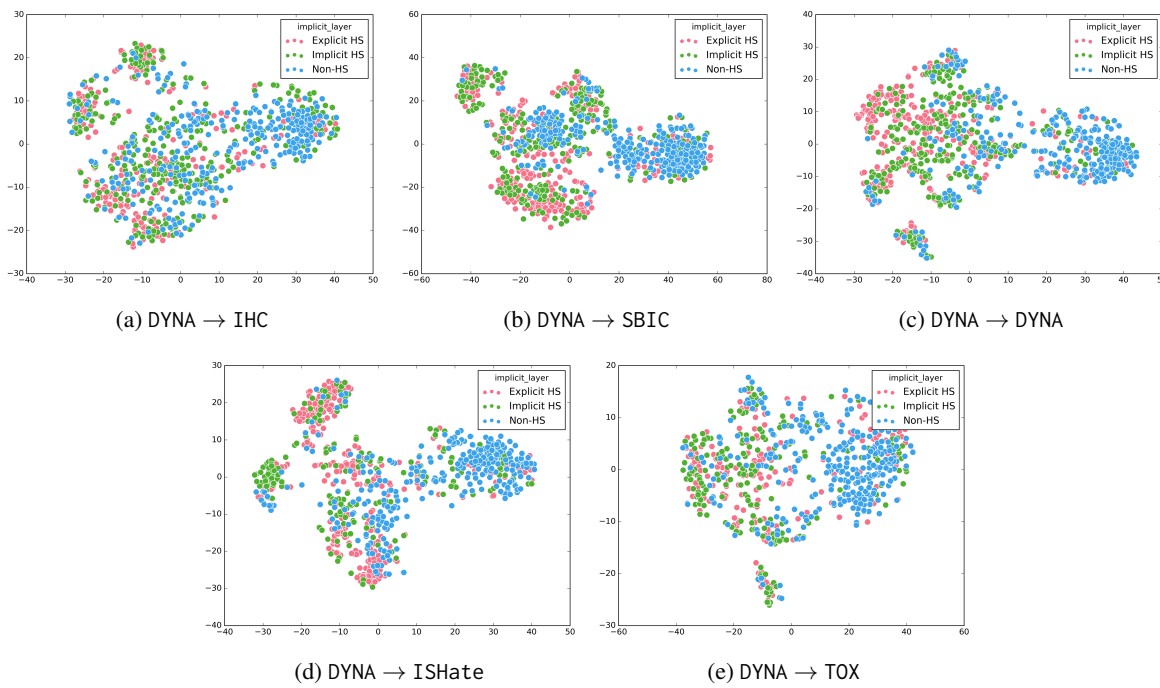

Figure 7: **RQ3**: TSNE embeddings of the test sets for all the datasets using HateBERT fine-tuned on DYNA and linking explicit and implicit instances

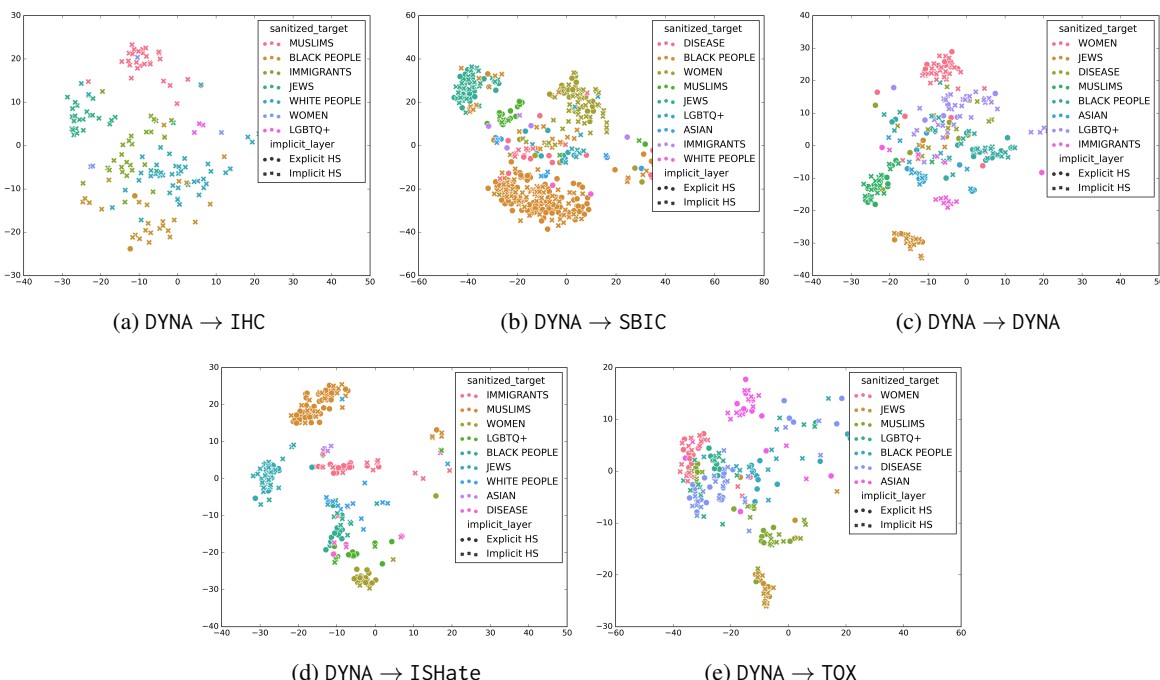

Figure 8: **RQ3**: TSNE embeddings of the test sets for all the datasets using HateBERT fine-tuned on DYNA based on their target groups. Non-hateful instances are excluded from these plots.