# OpenReview forum: "Unmasking the Hidden Meaning: Bridging Implicit and Explicit Hate Speech Embedding Representations"
_EMNLP/2023/Conference — EMNLP 2023 Findings_

### Official Review · Reviewer_QeTe · 2023-08-01

**Typos Grammar Style And Presentation Improvements:** 1) Line 133
**Soundness:** 3

**Excitement:**

3: Ambivalent: It has merits (e.g., it reports state-of-the-art results, the idea is nice), but there are key weaknesses (e.g., it describes incremental work), and it can significantly benefit from another round of revision. However, I won't object to accepting it if my co-reviewers champion it.

**Paper Topic And Main Contributions:**

The paper describes how implicit and explicit hate expressions affect hate speech detection. A cross-validation evaluation on different hate speech datasets shows that the models do not necessarily generalise across the datasets. Contrastive learning is employed to link implicit and explicit hate instances. Visualisations showing embedding representations show that contrastive learning may work well.

Contribution: The idea of using contrastive learning to link explicit and implicit hate is an interesting idea for hate speech detection.

**Questions For The Authors:**

1) Line 212: How is the loss function modified for contrastive learning? A formula may help.

2) Table 1: What do the numbers in bold face indicate?

3) While the visualisation after contrastive learning is interesting, it is not clear if it results in an improvement in hate speech detection. Do the authors have results around that?

**Reasons To Accept:**

1) The paper is well-written, and lays out the research questions systematically (however, there is a question about RQ1).

2) The problem of implicit hate messages is handled using contrastive learning. This seems to be an interesting use of contrastive learning.


**Reasons To Reject:**

1) It is not clear how the cross-evaluation results address RQ1.

2) It is not clear if contrastive learning improves hate speech classification - or generalisation across the datasets. The paper does not present results to suggest so.

**Reproducibility:**

3: Could reproduce the results with some difficulty. The settings of parameters are underspecified or subjectively determined; the training/evaluation data are not widely available.

**Reviewer Confidence:**

3: Pretty sure, but there's a chance I missed something. Although I have a good feel for this area in general, I did not carefully check the paper's details, e.g., the math, experimental design, or novelty.

---

> ### Author Rebuttal · Authors · 2023-08-28
>
> Thanks for your insightful comments.
>
> **1)** RQ1 investigates how machine learning models capture HS classes and whether explicit and implicit hateful messages are encoded differently across datasets. To answer this, we combine cross-evaluation with t-SNE analysis, focusing on HS, Non-HS, Explicit HS, and Implicit HS categories.
> The cross-evaluation, which involves training models on multiple HS datasets, provides us insights into the models' capabilities to recognize HS across various contexts and environments. This establishes a baseline understanding of the models' performance. The t-SNE plots visually demonstrate the impact of implicit and explicit factors on these embeddings. Together, these analyses offer a comprehensive view, satisfying the queries raised in RQ1.
>
> **2)** Regarding the paper's novelty, our work i) examines the variability of explicit and implicit hate speech (HS) across different target groups using five benchmark datasets, ii) we propose a method that bridges explicit and implicit HS in the representation space through contrastive learning techniques. We analyze those representations both before and after applying our technique. To strengthen the conclusions of our work, we conducted a supplemental experiment fine-tuning both BERT and HateBERT using our enhanced embeddings, adhering to the settings of our initial RQs by using multiple seeds and averaging accuracy on HateCheck (see table below). We show that the enhanced models **(CONT columns)** are better at correctly classifying challenging Non-HS messages across the five datasets. They also notably reduce false positives in HateCheck categories like quoted announcements (`counter_quote_nh`), direct references (`counter_ref_nh`), positive identifiers (`ident_pos_nh`), negated hateful remarks (`negate_neg_nh`), non-hateful profanity (`profanity_nh`), reclaimed slurs (`slur_reclaimed_nh`), homonym slurs (`slur_homonym_nh`) and targeted abuse at individuals (`target_indiv_nh`), objects (`target_obj_nh`), and other non-protected groups (`target_group_nh `). As a result, we reduce biases in non-hateful implicit cases often misclassified as Explicit HS due to trigger words or nuanced content. Our enhanced method maintains high performance on HS labels while improving classification in borderline cases, affirming the robustness and precision of our approach.
>
> We will include this new table as well as results on the cross-evaluation for BERT and HateBERT for Explicit, Implicit, and Non-HS categories in the camera-ready version if the paper gets accepted.
>
> | Test Case   	  | HateBERT IHC | CONT IHC | HateBERT SBIC | CONT SBIC | HateBERT DynaHate | CONT DynaHate | HateBERT ISHate | CONT ISHate | HateBERT ToxiGen | CONT ToxiGen |
> |-------------------|--------------|----------|---------------|-----------|--------------------|---------------|-----------------|-------------|------------------|--------------|
> | counter_quote_nh  | 0,486    	| 0,499	| 0,276     	| 0,422 	| 0,857          	| 0,962     	| 0,379       	| 0,524    	| 0,000        	| 0,010     	|
> | counter_ref_nh	| 0,576    	| 0,617	| 0,240     	| 0,393 	| 0,882          	| 0,902     	| 0,387       	| 0,521    	| 0,177        	| 0,295     	|
> | ident_pos_nh      | 0,441    	| 0,446	| 0,301     	| 0,361 	| 0,849          	| 0,767     	| 0,272       	| 0,350    	| 0,470        	| 0,575     	|
> | negate_neg_nh 	| 0,502    	| 0,517	| 0,120     	| 0,161 	| 0,448          	| 0,486     	| 0,164       	| 0,211    	| 0,087        	| 0,194     	|
> | profanity_nh  	| 0,796    	| 0,774	| 0,922     	| 0,994 	| 1,000          	| 1,000     	| 0,992       	| 0,996    	| 0,292        	| 0,456     	|
> | slur_homonym_nh   | 0,353    	| 0,413	| 0,393     	| 0,513 	| 0,813          	| 0,787     	| 0,680       	| 0,827    	| 0,320        	| 0,473     	|
> | slur_reclaimed_nh | 0,217    	| 0,277	| 0,472     	| 0,711 	| 0,891          	| 0,879     	| 0,741       	| 0,802    	| 0,272        	| 0,346     	|
> | target_group_nh   | 0,710    	| 0,700	| 0,623     	| 0,810 	| 0,968          	| 0,971     	| 0,448       	| 0,561    	| 0,000        	| 0,006     	|
> | target_indiv_nh   | 0,538    	| 0,572	| 0,655     	| 0,951 	| 1,000          	| 1,000     	| 0,782       	| 0,809    	| 0,003        	| 0,003     	|
> | target_obj_nh 	| 0,637    	| 0,622	| 0,923     	| 0,966 	| 0,969          	| 0,985     	| 0,735       	| 0,757    	| 0,006        	| 0,034     	|
>
> **Q1)** The contrastive loss operates on the embeddings corresponding to the CLS tokens in language models. It is defined as follows:
>
> ```python
> loss_cont = mean((1 - label_pair) * (similarity ** 2) + label_pair * (max(0, margin - similarity) ** 2))
> ```
>
> Here, `label_pair` is set to 1 for positive pairs and 0 for negative pairs. `similarity` refers to the cosine similarity calculated between a pair of messages, while `margin` serves as a hyperparameter that can be adjusted for optimization.
> For classification, we use the cross-entropy loss:
>
> ```python
> loss_clf = -sum(gold_label[i] * math.log(pred[i]) + (1 - gold_label[i]) * math.log(1 - pred[i]) for i in range(len(gold_label)))
> ```
>
> In this case, `gold_label` corresponds to the labels of the dataset on which the model is fine-tuned.
> The final loss function used for optimization during training is the sum of these two losses:
>
> ```python
> Total_loss = loss_cont + loss_clf
> ```
>
> By combining these losses, we optimize both the model's understanding of the embedding space and its classification performance.
>
> **Q2)** The table displays for each column, i.e. for each dataset  the top two scores. As could be expected, the highest score is obtained when training and testing on the same dataset, serving as a best-case benchmark. The second-highest score reveals the next most effective model.
>
> **Q3)** It is answered in points 1) and 2).

---

### Official Review · Reviewer_UB9Q · 2023-08-01

**Soundness:** 4

**Excitement:**

3: Ambivalent: It has merits (e.g., it reports state-of-the-art results, the idea is nice), but there are key weaknesses (e.g., it describes incremental work), and it can significantly benefit from another round of revision. However, I won't object to accepting it if my co-reviewers champion it.

**Missing References:**

Reference - https://aclanthology.org/2023.findings-eacl.9/ (already mentioned in review above)

**Paper Topic And Main Contributions:**

This paper studies the relation between implicit and explicit hate-speech detection. They study the learned embedding distribution for both cases as well showing they are pivoted around target groups visualised using t-sne plots. They employ an contrastive learning strategy to bring together the the explicit and implicit embeddings based on target groups.

**Reasons To Accept:**

1) The authors uncover an important insight that explicit and implicit hate speech embeddings are anchored around target groups.
2) They also show that using a contrastive learning strategy the above mentioned embeddings can be brought even closer to each other.

**Reasons To Reject:**

1) The RQ1 mentioned in the paper seems redundant. This adds no extra information for the audience. It is expected the performance will vary across multiple HS datasets  when evaluated in cross-data setting. Another interesting point to analyse would've been how % of explicit hate information in the dataset affects implicit hate speech detection performance and vice-versa and it's corresponding effect on RQ2 & RQ3 t-sne plots. (Reference - https://aclanthology.org/2023.findings-eacl.9/)

2) Again it is only obvious / intuitive that employing a contrastive learning strategy would bring together the implicit and explicit hate embeddings. What would be interesting to understand is that how can these correlations be leveraged to improve the downstream classification performance ?

In it's current form the paper lacks enough significant learnings / contribution to be accepted. Incorporating the above mentioned suggestions should be sufficient

**Reproducibility:**

4: Could mostly reproduce the results, but there may be some variation because of sample variance or minor variations in their interpretation of the protocol or method.

**Reviewer Confidence:**

5: Positive that my evaluation is correct. I read the paper very carefully and I am very familiar with related work.

---

> ### Author Rebuttal · Authors · 2023-08-28
>
> Thanks for your insightful comments.
>
> **1)** In response to RQ1 appearing redundant, our aim was not just to show that performance varies in a cross-data setting but to quantify the extent of this variation. Thanks for the suggested reference. We will cite it in the paper. Following that, and to strengthen the conclusions of our work, we conducted a supplemental experiment fine-tuning both BERT and HateBERT using our enhanced embeddings, adhering to the settings of our initial RQs by using multiple seeds and averaging accuracy on HateCheck (see table below). We show that the enhanced models **(CONT columns)** are better at correctly classifying challenging Non-HS messages across the five datasets. They also notably reduce false positives in HateCheck categories like quoted announcements (`counter_quote_nh`), direct references (`counter_ref_nh`), positive identifiers (`ident_pos_nh`), negated hateful remarks (`negate_neg_nh`), non-hateful profanity (`profanity_nh`), reclaimed slurs (`slur_reclaimed_nh`), homonym slurs (`slur_homonym_nh`) and targeted abuse at individuals (`target_indiv_nh`), objects (`target_obj_nh`), and other non-protected groups (`target_group_nh `). As a result, we reduce biases in non-hateful implicit cases often misclassified as Explicit HS due to trigger words or nuanced content. Our enhanced method maintains high performance on HS labels while improving classification in borderline cases, affirming the robustness and precision of our approach.
> | Test Case   	  | HateBERT IHC | CONT IHC | HateBERT SBIC | CONT SBIC | HateBERT DynaHate | CONT DynaHate | HateBERT ISHate | CONT ISHate | HateBERT ToxiGen | CONT ToxiGen |
> |-------------------|--------------|----------|---------------|-----------|--------------------|---------------|-----------------|-------------|------------------|--------------|
> | counter_quote_nh  | 0,486    	| 0,499	| 0,276     	| 0,422 	| 0,857          	| 0,962     	| 0,379       	| 0,524    	| 0,000        	| 0,010     	|
> | counter_ref_nh	| 0,576    	| 0,617	| 0,240     	| 0,393 	| 0,882          	| 0,902     	| 0,387       	| 0,521    	| 0,177        	| 0,295     	|
> | ident_pos_nh      | 0,441    	| 0,446	| 0,301     	| 0,361 	| 0,849          	| 0,767     	| 0,272       	| 0,350    	| 0,470        	| 0,575     	|
> | negate_neg_nh 	| 0,502    	| 0,517	| 0,120     	| 0,161 	| 0,448          	| 0,486     	| 0,164       	| 0,211    	| 0,087        	| 0,194     	|
> | profanity_nh  	| 0,796    	| 0,774	| 0,922     	| 0,994 	| 1,000          	| 1,000     	| 0,992       	| 0,996    	| 0,292        	| 0,456     	|
> | slur_homonym_nh   | 0,353    	| 0,413	| 0,393     	| 0,513 	| 0,813          	| 0,787     	| 0,680       	| 0,827    	| 0,320        	| 0,473     	|
> | slur_reclaimed_nh | 0,217    	| 0,277	| 0,472     	| 0,711 	| 0,891          	| 0,879     	| 0,741       	| 0,802    	| 0,272        	| 0,346     	|
> | target_group_nh   | 0,710    	| 0,700	| 0,623     	| 0,810 	| 0,968          	| 0,971     	| 0,448       	| 0,561    	| 0,000        	| 0,006     	|
> | target_indiv_nh   | 0,538    	| 0,572	| 0,655     	| 0,951 	| 1,000          	| 1,000     	| 0,782       	| 0,809    	| 0,003        	| 0,003     	|
> | target_obj_nh 	| 0,637    	| 0,622	| 0,923     	| 0,966 	| 0,969          	| 0,985     	| 0,735       	| 0,757    	| 0,006        	| 0,034     	|
>
> Additionally, we evaluated the influence of both explicit and implicit forms of hate speech on the detection capabilities of our enhanced HateBERT model. The table below provides a snapshot of the model's average accuracy across three pivotal categories defined on the DynaHate test set: Explicit, Implicit, and Non-HS. These results are calculated as the mean across five model instances for each training dataset. Model instances are initiated with a unique random seed.
>
> | Train 	| Explicit | Implicit | Non-HS  |
> |-----------|----------|----------|---------|
> | IHC   	| 0,6855   | 0,8184   | 0,3447  |
> | SBIC  	| 0,6566   | 0,6592   | 0,6638  |
> | DynaHate  | 0,8187   | 0,7186   | 0,8117  |
> | ISHate	| 0,7326   | 0,5870   | 0,5878  |
> | ToxiGen   | 0,8353   | 0,8061   | 0,2708  |
>
> The data illustrates that Explicit and Implicit categories demonstrate comparably high levels of accuracy in hate speech detection. This underscores their near-equivalent influence on the model's overall performance. Equally important is the significance of the Non-HS category, which varies across different training data sets but remains an essential factor for comprehensive hate speech detection.
>
> We will include these new tables as well as results on the cross-evaluation for BERT and HateBERT for Explicit, Implicit, and Non-HS categories in the camera-ready version if the paper gets accepted.
>
> **2)** In response to the comment regarding the use of contrastive learning to bring implicit and explicit hate embeddings closer, we've taken it a step further. We've innovated our training optimizer to compute cosine similarities between positive and negative pairs of embeddings, which are based on the CLS tokens in language models. Positive pairs are drawn closer in the representation space, while negative pairs are pushed apart. These refined embeddings are then channeled to a classification layer. Our approach highlights the unique combination of cross-entropy loss with contrastive loss. This dual optimization strategy allows for effective training of downstream classification tasks, integrating both contrastive learning and cross-entropy in a seamless manner. The effectiveness of this novel architecture is evidenced by the results highlighted in **1)**.

---

### Official Review · Reviewer_TtZW · 2023-08-04

**Soundness:** 3

**Excitement:**

3: Ambivalent: It has merits (e.g., it reports state-of-the-art results, the idea is nice), but there are key weaknesses (e.g., it describes incremental work), and it can significantly benefit from another round of revision. However, I won't object to accepting it if my co-reviewers champion it.

**Paper Topic And Main Contributions:**

The paper discusses the problem of hate speech from a computational perspective, which is motivated by practical scenarios where hate speech on social media often appears in implicit forms and automated detection is necessary. In exploring the impact of explicitness and target groups on hate speech detection, the paper presents visualizations of hate speech embeddings and proposes new ones using contrastive learning. Through these visualizations, the paper reveals the learned representations of hate speech across different groups and levels of explicitness, while using contrastive learning to enhance the representations of implicit hate speech.

**Reasons To Accept:**

The paper is well-structured, and the figures included are illustrative.
Besides, the paper conducts extensive experiments on five datasets, and demonstrates an adequate understanding of hate speech detection.

**Reasons To Reject:**

Although hate speech detection has been a topic of concern for some time, this paper does not significantly contribute to the field. There is little novelty in either methodology or conclusion. Specifically, the paper primarily utilizes existing visualization tools (TSNE) and pre-trained models (BERT and HateBERT). The proposed method for RQ3 resembles a pipeline rather than a novel model architecture. Furthermore, the findings are limited, with no further analysis of linguistic or causal mechanisms, providing little implications for research, practice, or society. Therefore, the paper seems more like a visualized experiment report rather than a linguistic analysis or modeling work.
Moreover, the conclusions drawn from the experimental results are weak, with some lacking supporting evidence. For instance, in section 3.4, the paper claims that the embeddings of explicit hate speech and non-hate speech are clearly separated, but figure 1 indicates that pink and blue dots actually mix to a large extent. Similarly, in figure 4, while the embeddings of explicit and implicit hate speech are pulled closer as expected, the distance between the embeddings of hate speech and non-hate speech remains unclear. This raises doubts about the effectiveness of the proposed method in practical hate speech detection scenarios.
While some of the research questions are intriguing, the analysis presented in the paper is not sufficiently thorough. For instance, regarding the target group of hate speech, it is suggested to conduct quantitative research on the explicitness of such speech and further analyze the similarities and differences among them.
Additionally, using different datasets with varying standards for implicit hate speech labeling (or even hate speech labeling) may lead to inconsistencies, potentially causing reduced accuracy and reliability in the results.

**Reproducibility:**

4: Could mostly reproduce the results, but there may be some variation because of sample variance or minor variations in their interpretation of the protocol or method.

**Reviewer Confidence:**

4: Quite sure. I tried to check the important points carefully. It's unlikely, though conceivable, that I missed something that should affect my ratings.

**Typos Grammar Style And Presentation Improvements:**

When using an abbreviation for the first time in the paper, it is recommended to provide its full name as well.

---

> ### Author Rebuttal · Authors · 2023-08-28
>
> Thanks for your insightful comments.
>
> **1)** Regarding the paper's novelty, our work i) examines the variability of explicit and implicit hate speech (HS) across different target groups using five benchmark datasets, ii) we propose a method that bridges explicit and implicit HS in the representation space through contrastive learning techniques. We analyze those representations both before and after applying our technique. To strengthen the conclusions of our work, we conducted a supplemental experiment fine-tuning both BERT and HateBERT using our enhanced embeddings, adhering to the settings of our initial RQs by using multiple seeds and averaging accuracy on HateCheck (see table below). We show that the enhanced models **(CONT columns)** are better at correctly classifying challenging Non-HS messages across the five datasets. They also notably reduce false positives in HateCheck categories like quoted announcements (`counter_quote_nh`), direct references (`counter_ref_nh`), positive identifiers (`ident_pos_nh`), negated hateful remarks (`negate_neg_nh`), non-hateful profanity (`profanity_nh`), reclaimed slurs (`slur_reclaimed_nh`), homonym slurs (`slur_homonym_nh`) and targeted abuse at individuals (`target_indiv_nh`), objects (`target_obj_nh`), and other non-protected groups (`target_group_nh `). As a result, we reduce biases in non-hateful implicit cases often misclassified as Explicit HS due to trigger words or nuanced content. Our enhanced method maintains high performance on HS labels while improving classification in borderline cases, affirming the robustness and precision of our approach.
>
> We will include this new table as well as results on the cross-evaluation for BERT and HateBERT for Explicit, Implicit, and Non-HS categories in the camera-ready version if the paper gets accepted.
>
> | Test Case		 | HateBERT IHC | CONT IHC | HateBERT SBIC | CONT SBIC | HateBERT DynaHate | CONT DynaHate | HateBERT ISHate | CONT ISHate | HateBERT ToxiGen | CONT ToxiGen |
> |-------------------|--------------|----------|---------------|-----------|--------------------|---------------|-----------------|-------------|------------------|--------------|
> | counter_quote_nh  | 0,486        | 0,499    | 0,276         | 0,422     | 0,857              | 0,962         | 0,379           | 0,524        | 0,000            | 0,010         |
> | counter_ref_nh    | 0,576        | 0,617    | 0,240         | 0,393     | 0,882              | 0,902         | 0,387           | 0,521        | 0,177            | 0,295         |
> | ident_pos_nh 	 | 0,441        | 0,446    | 0,301         | 0,361     | 0,849              | 0,767         | 0,272           | 0,350        | 0,470            | 0,575         |
> | negate_neg_nh     | 0,502        | 0,517    | 0,120         | 0,161     | 0,448              | 0,486         | 0,164           | 0,211        | 0,087            | 0,194         |
> | profanity_nh      | 0,796        | 0,774    | 0,922         | 0,994     | 1,000              | 1,000         | 0,992           | 0,996        | 0,292            | 0,456         |
> | slur_homonym_nh   | 0,353        | 0,413    | 0,393         | 0,513     | 0,813              | 0,787         | 0,680           | 0,827        | 0,320            | 0,473         |
> | slur_reclaimed_nh | 0,217        | 0,277    | 0,472         | 0,711     | 0,891              | 0,879         | 0,741           | 0,802        | 0,272            | 0,346         |
> | target_group_nh   | 0,710        | 0,700    | 0,623         | 0,810     | 0,968              | 0,971         | 0,448           | 0,561        | 0,000            | 0,006         |
> | target_indiv_nh   | 0,538        | 0,572    | 0,655         | 0,951     | 1,000              | 1,000         | 0,782           | 0,809        | 0,003            | 0,003         |
> | target_obj_nh     | 0,637        | 0,622    | 0,923         | 0,966     | 0,969              | 0,985         | 0,735           | 0,757        | 0,006            | 0,034         |
>
>
> **2)** Regarding RQ3 and the novel architecture. We modify the training optimizer to calculate cosine similarities between positive and negative pairs of embeddings corresponding to the CLS tokens in language models. Positive pairs are drawn together, while negative ones are pushed apart. These enhanced embeddings then go through a classification layer. Our training then combines cross-entropy loss with contrastive loss. The cross-entropy loss uses the gold labels from the datasets instead of the sample pairs. Hence, it is not merely a pipeline but a novel, integrated model architecture that influences learning, representation, and classification.
>
> **3)** Figure 1 illustrates the baseline results *without* our technique, which is why the pink and blue dots are substantially mixed. This shows the need for our novel, integrated approach, which aims to better segregate these categories in the representation space (see table above).
>
> **4)** Running experiments across varied datasets aimed to show how our experiments consistently support our claims across disparate contexts, topics, and social media. This suggests that our method is robust, has broader applicability, and can be effectively generalized.

---

### Meta-Review · Area_Chair_vmT2 · 2023-09-18

**Recommendation:** 3

**Metareview:**

This paper proposes the use of a contrastive learning based strategy to boost performance on hate speech detection with a focus on both implicit and explicit instances. Experiments show the overall potential of the proposed solution to improve performance over other baselines.

The experiments are overall sound and the problem tackled is an important and challenging one, with one of the weaker points being that the proposed solution is somewhat incremental. Another aspect that would have been good to see is some more depth in the experimentation, possibly with ablation experiments to demonstrate where the improvement is coming from, e.g. whether it is from the use of contrastive learning.

While this is somewhat understandable from a short paper, I appreciate the extensive rebuttal with new results addressing the above limitations and I would appreciate the authors to incorporate these in a further revision.

---

### Decision · Program_Chairs · 2023-10-07

**Decision:**

Accept-Findings

**Comment:**

This paper proposes the use of a contrastive learning based strategy to boost performance on hate speech detection with a focus on both implicit and explicit instances. Experiments show the overall potential of the proposed solution to improve performance over other baselines.

The experiments are overall sound and the problem tackled is an important and challenging one, with one of the weaker points being that the proposed solution is somewhat incremental. Another aspect that would have been good to see is some more depth in the experimentation, possibly with ablation experiments to demonstrate where the improvement is coming from, e.g. whether it is from the use of contrastive learning.

While this is somewhat understandable from a short paper, I appreciate the extensive rebuttal with new results addressing the above limitations and I would appreciate the authors to incorporate these in a further revision.